# CXCR5 Signals Fine-Tune Dendritic Cell Transcription and Regulate T_H_2 Development

**DOI:** 10.3390/vaccines13090943

**Published:** 2025-09-03

**Authors:** Miranda L. Curtiss, Natalia Ballesteros Benavides, Alexander F. Rosenberg, Christopher D. Scharer, Beatriz León, Frances E. Lund

**Affiliations:** 1Division of Pulmonary and Critical Care Medicine, Department of Medicine, The University of Alabama at Birmingham, Birmingham, AL 35294, USA; 2Department of Microbiology, The University of Alabama at Birmingham, Birmingham, AL 35294, USAbeatriz.leonruiz@nih.gov (B.L.); flund@uab.edu (F.E.L.); 3Department of Biomedical Informatics and Data Science, The University of Alabama at Birmingham, Birmingham, AL 35294, USA; afr@uab.edu; 4Department of Microbiology and Immunology, Emory University, Atlanta, GA 30322, USA; cdschar@emory.edu; 5Laboratory of Allergic Diseases, National Institute of Allergy and Infectious Diseases, National Institutes of Health, Bethesda, MD 20892, USA

**Keywords:** dendritic cells, T_H_2 responses, CXCR5, Chi3l1, helminth infection

## Abstract

Background/Objectives: We previously demonstrated that dendritic cell (DC) expression of CXCR5 is required for T_H_2 priming in mice infected with the helminth *Heligmosomoides polygyrus (Hp)*. In this manuscript we examined how CXCR5 controls DC mediated CD4 T helper 2 cell (T_H_2) development. Methods: We used *in vitro* T_H_2 priming assays, RNA-seq analyses and *in vivo Hp* infection mouse models to identify roles for the CXCR5-expressing DCs in T_H_2 development. Results: We showed that migratory conventional type 2 dendritic cells (cDC2) express CXCR5 and that deletion of *Cxcr5* prevents migratory DC priming of T_H_2 cells *in vitro* while overexpression of CXCR5 enhances migratory DC priming of T_H_2 cells *in vitro*. To understand how CXCR5 facilitates the T_H_2 priming capabilities of migratory cDC2 cells, we performed RNAseq analysis on wildtype and *Cxcr5*^−/−^ DC subsets isolated from msLN of *Hp*-infected mice. We observed that CXCR5 expression specifically by the migratory cDC2 subset promoted a pro-proliferative transcriptional program in cDC2 cells and was required for cDC2 cell accumulation in the msLN following *Hp* infection. We demonstrated that CXCR5 expression specifically by cDC2 cells was necessary for upregulation of *Chitinase 3-like-1 (Chi3l1)*, which encodes a secreted protein (Chi3l1) that regulates allergic T_H_2 responses. We showed that addition of recombinant Chi3l1 protein to *in vitro* T_H_2 priming cultures enhanced T_H_2 development and that deletion of *Chi3l1* specifically in DCs resulted in fewer cDC2 cells and decreased T_H_2 development *in vivo* following *Hp* infection. Conclusions: CXCR5 expressed by cDC2 cells is required for induction of *Chi3l1*, which in turn promotes the T_H_2 priming capacity of these DCs. These findings provide insight into the actions of CXCR5 and Chi3l1 in helminth infection.

## 1. Introduction

Dendritic cells (DCs) are professional antigen-presenting cells that can take up antigen, process antigen and display peptides from the processed antigen to CD4 and CD8 T cells to facilitate T cell activation or, in some settings, T cell tolerance [1]. In addition to presenting antigens, DCs express co-stimulatory receptors and secreted mediators that support differentiation of naïve CD4 T cells into specialized T helper cell subsets, including T_H_2 cells [2]. Dendritic cells comprise the ontologically distinct lineages of skin-resident Langerhans cells, plasmacytoid dendritic cells that respond to viral infections, and conventional dendritic cells (cDC) that are recruited to inflamed tissues during enteric helminth infections, migrate to draining lymph nodes and promote T_H_2 cell development [1,2,3]. Lymph node cDC subsets include resident cDCs (CD11c^hi^MHCII^+^) that play a primarily homeostatic role and migratory cDCs (CD11c^+^MHCII^hi^) that emigrate from inflamed or infected tissue to lymph nodes (LN) to initiate antigen-specific T cell responses [2]. Murine migratory DCs include the well-established cDC1 (CD11b^neg^CD103^+^) and cDC2 (CD11b^+^CD103^neg^) subsets [4,5] as well as the less well-studied DP (CD103^+^CD11b^+^) and DN (CD103^neg^CD11b^neg^) populations. cDC2 cells are known to regulate CD4 T cell priming [1], while cDC1 cells play regulatory roles and support cross-priming of CD8 T cells [6]. Although less is known about the functional attributes of DP and DN DC subsets during infection, both populations are reported to be part of the broader cDC2 lineage and are distinct from cDC1 cells [7,8,9,10]. cDC2, DP and DN cDCs can promote T_H_2 responses in murine infection and allergic disease models [5,7,11,12]. Not surprisingly, distinct transcription factors can be used to define migratory cDC1 and cDC2 cells [1,13] with IRF4 required for DN cell development [11] and for cDC2 function [14] while IRF8 and BATF3 are required for cDC1 cell development and function [8,15,16].

Murine models of helminth infection reveal important roles for the cDC2 subset during T_H_2 cell priming. For example, DCs from *Heligmosomoides polygyrus (Hp)*-infected *Batf3*^−/−^ mice, which retain cDC2 cells and lack cDC1 cells [15,17], manifest enhanced T_H_2 responses after *Hp* infection [8]. This is likely because the *Batf3*^−/−^ DCs fail to express IL-12 and IL-12 is well known to inhibit T_H_2 polarization [1]. IRF4, on the other hand, is important for preventing IL-12 expression by cDC2 cells in *Leishmania major*-infected mice [9]. Likewise, conditional deletion of *Irf4* in DCs blocks cDC2 development and impairs T_H_2 responses to infection with *Nippostrongylus brasiliensis (Nb)* [14]. Similar results were reported using mouse models of allergic inflammation and asthma [14,18]. Moreover, DC specific deletion of *Klf4*—a transcription factor downstream of IRF4—inhibits *in vivo* T_H_2 responses to allergens and *Schistosoma mansonii* [7]. *In vitro* CD4 T cell priming assays and adoptive transfer experiments provide further evidence that cDC2 cells are directly required for T_H_2 development. Indeed, cDC2 cells, which were isolated from mesenteric lymph nodes (msLN) of *Hp*-infected mice and then cultured with naïve CD4 T cells *in vitro,* spontaneously primed T_H_2 development in the absence of exogenous IL-4 [19]. Likewise, adoptive transfer of cDC2 cells from *Nb*-infected mice into naïve hosts is sufficient to support T_H_2 priming *in vivo* [20]. Collectively, these data argue that induction of protective T_H_2 responses following helminth infection is dependent on the migratory cDC2 subset.

Prior work from multiple labs [3,21,22,23,24,25] show that migratory cDC1 and cDC2 cells localize to distinct areas of the LN. Migratory cDC1 cells as well as resident DCs express high levels of CCR7 and preferentially migrate to the T cell zone of the lymph node [3] in response to the CCR7 ligands, CCL19 and CCL21, which are produced by specialized fibroblastic reticular cells (FRCs) that reside in the T cell zone [21,22,23]. On the other hand, we showed that migratory DCs elicited in *Hp*-infected mice express lower levels of CCR7 and have upregulated expression of CXCR5 [3]. These CXCR5^+^ DCs localize to the interfollicular or perifollicular region of the LN near the T-B border where the CXCR5 ligand, CXCL13, is expressed by stromal cells [24,25]. Separation of the CXCR5-expressing migratory DCs from the T cell zone of the lymph nodes appears important for the function of these cells as deletion of *Cxcr5* in DCs impairs development of IL-4^+^ CD4 T cell responses during both *Hp* [3] and *Trichuris muris* [22] infections. Thus, CXCR5 expression by migratory DCs is important for development of protective T_H_2 immunity during helminth infection.

Our prior data showed that expression of CXCR5 by both DCs and CD4 T cells is important for the development of a robust T_H_2 response to *Hp* infection [3]. While it was possible that CXCR5 expression by the migratory DCs is only necessary to co-localize the DCs with the T_H_2 precursors within the perifollicular region of the LN, we hypothesized that the migratory cDC2 cells might also receive signals within this specialized microenvironment that conditioned these DCs to support T_H_2 development. Consistent with this hypothesis, we found that CXCR5 was expressed by cDC2 cells but not by cDC1 cells. To directly test this hypothesis, we examined the *in vitro* T_H_2 priming capacity of wildtype, *Cxcr5*^−/−^ and CXCR5 over-expressing migratory DCs. We showed that deletion of *Cxcr5* in migratory DCs prevented *in vitro* T_H_2 priming of naïve T cells while overexpression of CXCR5 by migratory DCs enhanced T_H_2 priming. These results therefore suggested that DC intrinsic expression of CXCR5 was not simply necessary to co-localize DCs and T cells but was also necessary for the functional attributes of the DCs. To further analyze this, we undertook an unbiased RNA sequencing approach to characterize the CXCR5-regulated genes expressed by msLN cDC2 cells during *Hp* infection. We found that a small set of genes were differentially expressed in *Cxcr5*^−/−^ cDC2 cells, including genes that support proliferation and homeostasis of the cDC2 subset in the msLN. In addition, we found that CXCR5 controls expression of *Chi3l1*, a gene that encodes a secreted protein known to regulate T_H_2 immunity [26]. Addition of recombinant Chi3l1 protein to *in vitro* DC/T_H_2 priming cultures enhanced T_H_2 development. Using a mouse model in which *Chi3l1* was selectively deleted in DCs, we demonstrated that T_H_2 responses were significantly attenuated following *Hp* infection. Therefore, we conclude that CXCR5 expression by the T_H_2 priming cDC2 subset supports upregulation of *Chi3l1*, and Chi3l1 can, in turn, function as a DC-derived secretory factor to support protective T_H_2 responses following infection.

## 2. Materials and Methods

### 2.1. Mice

All experimental animals were bred and maintained in the University of Alabama at Birmingham (UAB) animal facilities. All procedures involving animals were approved by the UAB Institutional Animal Care and Use Committee and were conducted in accordance with the principles outlined by the National Research Council. CD45.2^+^ C57BL/6J (B6 mice), CD45.1^+^ B6.SJL-Ptprc^a^ Pepc^b^/BoyJ (CD45.1^+^ B6 Pepboy mice [27,28]), B6.129S2(Cg)-*Cxcr5*^tm1Lipp^/J (*Cxcr5*^−/−^ mice [29]), B6.Cg-Tg(TcraTcrb)425Cbn/J (OT-II^Tg^ mice [30]), B6.Cg-Tg(Itgax-cre)1-1Reiz/J (CD11c-cre mice [31]), C.FVB-1700016L21Rik^Tg(Itgax-DTR/EGFP)57Lan^/J (CD11c-DTR^Tg/−^ mice [32]), BALB/cByJ (BALB/c mice), and CBy.PL(B6)-Thy1^a^/ScrJ (BALB/c.Thy1.1^+/+^ mice [33]) were purchased from the Jackson Laboratories (JAX, Bar Harbor, ME, USA). BRP-39 deficient mice (*Chi3l1*^−/−^ mice [26]) on the BALB/c genetic background were provided as a kind gift by Dr. Allison Humbles (MedImmune). CD11c-DTR^Tg/−^.Thy1.1^+/+^ mice on the BALB/c genetic background were produced at UAB by backcrossing CD11c-DTR^Tg/−^ mice on the B6 genetic background to BALB/c mice for 10 generations. These mice were then intercrossed with BALB/c Thy1.1^+/+^ mice for an additional 6 generations. CD45.1^+^OT-II^Tg^.4get^Tg/−^ mice on the B6 genetic background (referred to as “OT-II 4get”) were a kind gift of Dr. B. Leon. This strain was produced by first intercrossing OT-II^Tg/−^ mice with CD45.1^+^ B6 Pepboy mice to generate OT2^Tg/−^.CD45.1^+/+^ mice. B6.OT-II^Tg^.4get^Tg/−^CD45.1^+/−^ mice were then produced by crossing B6.OTII^Tg^.CD45.1^+/+^ male mice to 4get^Tg/Tg^ female mice on the B6 genetic background (the B6.4get^Tg/Tg^ mice were produced by intercrossing BALB/c 4get^Tg^ mice (C.129-Il4^tm1Lky^/J, JAX) to B6 mice for 10 generations). Male mice carrying the OT-II transgene on the Y chromosome were used for experiments. *CD11c-Cxcr5^Tg^* mice on the B6 genetic background were generated by the UAB transgenic mouse facility. Briefly, murine B6 ES cells were transfected with a ROSA26 locus targeting vector containing a loxP-STOP-loxP cassette followed by a murine *Cxcr5* cDNA-IRES-EGFP expression cassette. ES cells containing the transgene knocked into the ROSA26 locus were injected into B6 blastocytes and offspring containing the knocked in allele were selected and interbred with CD11c-cre mice. Animals expressing both transgenes were used in experiments.

### 2.2. Bone Marrow Chimeras

Bone marrow (BM) chimeric mice were generated by irradiating BALB/c or B6 CD45.1^+^ Pepboy recipient animals with 850 Rads from a high-energy X-ray source, delivered in a split dose 5 h apart, and then reconstituting the recipients with 5 × 10^6^ BM cells by retro-orbital injection. To generate animals lacking *Chi3l1* specifically in DCs, we produced BM chimeras by reconstituting irradiated BALB/c mice with 4 × 10^6^ CD11c.DTR^Tg/−^.Thy1.1^+/+^ BM + 1 × 10^6^ BALB/c BM (DTR-WT chimeras) or 4 × 10^6^ CD11c. DTR^Tg/−^.Thy1.1^+/+^ BM + 1 × 10^6^ *Chi3l1*^−/−^ BM (DTR-*Chi3l1*^−/−^ chimeras). To generate 1:1 B6:*Cxcr5*^−/−^ chimeras we reconstituted irradiated B6 CD45.1^+^ Pepboy mice with 2.5 × 10^6^ CD45.1^+^ B6 Pepboy BM + 2.5 × 10^6^ *Cxcr5*^−/−^ BM. BM chimeras were used in experiments at 8–12 weeks post-reconstitution. Both male and female mice were used in this study. Within each experiment, animals were matched for age, 8–12 weeks at time zero, and sex. No differences were observed between cohorts of male versus female mice.

### 2.3. Infections and Diphtheria Toxin Treatment

*H polygyrus (Hp)* was maintained as described previously [3,34]. For infections, mice were gavaged with 200 *H polygyrus* L3 larvae. In some experiments 100 ng diphtheria toxin (DT, Sigma-Aldrich, St. Louis, MO, USA) was administered *i.p.* the day of gavage with *Hp* and repeated every 48 h.

### 2.4. Flow Cytometry

Mice were sacrificed at 8 or 14 days post-infection (*p.i.*) for analysis of mesenteric lymph node (msLN) cells during acute *Hp* infection. Single cell suspensions from msLN or spleens (spleens were first subjected to red blood cell lysis with ACK lysis buffer (0.15 M NH_4_Cl, 10 mM KHCO_3_, 0.1 mM EDTA) were preincubated with Fc receptor blocking antibody anti-CD16/32 (2.4G2 BioXCell, Lebanon, NH, USA), then stained with cocktails of labeled antibodies. Antibody clone names and vendor include the following: CD3 (17A2 BD), CD4 (GK1.5, BD and Biolegend, San Diego, CA, USA), CD11b (M1/70 BD), CD11c (HL3 BD, N418 Biolegend and eBioscience, San Diego, CA, USA), CD19 (1D3 eBioscience), CD44 (IM7 BD Horizon), CD45.1 (A20 BD, eBioscience and Biolegend), CD45.2 (104 BD and eBioscience), B220 (RA3-6B2 BD and eBioscience), CD64 (X54-5/7.1 Biolegend, 290,322 R&D Systems), CD49b (DX5 Biolegend), CD90.1/Thy1.1 (HIS51 eBiosciences), CD90.2/Thy1.2 (53-2.1 BD), CD103 (M290 BD Biosciences, 2E7 eBioscience), CD185/CXCR5 (2G8 BD Biosciences), NK1.1 (PK136 BD, BioLegend and eBioscience), GR-1 (RB6-8C5 Biolegend), Ly6C (AL-21 BD), I-A[b] (Af6-120.1 BD), I-A[d] (AMS-32.1 BD), I-A/I-E (M5/114.15.2 Biolegend). Other reagents include streptavidin (eBioscience), 7AAD (Millipore-Sigma, Burlington, MA, USA) and LIVE/DEAD^®^ red (Life Technologies, Waltham, MA, USA).

To determine the number and phenotypic characteristics of the DCs, msLNs of uninfected and *Hp*-infected mice were subjected to mechanical disruption with glass slides, then blocked with 2.4G2 antibody. msLN cells were stained with antibodies to I-A/I-E or I-A[b] (B6 genetic background), I-A/I-E or I-A[d] (BALB/c genetic background), CD11c, CXCR5, CD11b, CD103, and a lineage dump panel (7AAD, CD3, B220, GR-1, CD64, plus NK1.1 (B6 genetic background) or DX5 (BALB/c genetic background).

To detect intracellular cytokines, msLN cells were restimulated with 2.5 μg/mL plate-bound anti-CD3 (145-2C11, BioXCell) in the presence of 12.5 μg/mL brefeldin A (Sigma) for 4 h, blocked with Fc block (2.4G2 BioXCell), and then surface stained (CD4, CD44, Thy1.1 and Thy1.2). After washing, the cells were fixed in formalin, permeabilized with 0.1% NP-40, washed, incubated with the anti-cytokine antibodies IL-4 (11B11, BD and Invitrogen, Waltham, MA, USA) and IL-13 (eBio13A, eBioscience) in permeabilization buffer and washed. All incubations prior to fixation were performed in the presence of brefeldin A. All flow analysis was performed on a BD Canto.

### 2.5. DC Enrichment and Sort Purification

For RNA-seq, chimeras were generated from B6 Pepboy CD45.1^+^ mice irradiated and reconstituted with 50% B6 Pepboy CD45.1^+^ BM and 50% *Cxcr5*^−/−^ CD45.2^+^ BM (1:1 B6:*Cxcr5*^−/−^). A total of 8 days post-infection (*p.i.*) with 200 *Hp* L3 larvae, msLN were digested with 0.6 mg/mL collagenase A (Sigma) and 30 μg/mL DNase I (Sigma) in RPMI-1640 medium (GIBCO, Waltham, MA, USA) for 30 min, then mechanically disrupted with glass slides. DCs were positively enriched with anti-CD11c microbeads (Miltenyi, San Diego, CA, USA) in the presence of Fc blocking antibody (anti-CD16/32; 2.4G2 BioXCell), then stained with fluorochrome-conjugated antibodies to MHC II, CD11c, CD45.1, CD45.2, CD19, CCR7, CXCR5, CD11b, and CD103. CD11c bead-enriched cells were sorted using a BD Aria (UAB Flow Cytometry and Single Cell Core (FCSC Core)) into viable CD11c^+^MHCII^hi^CD19^−^ cells and further subdivided to obtain CD45.1^+^ (B6) and CD45.2^+^ (*Cxcr5*^−/−^) cDC2 (CD11b^+^CD103^−^) and cDC1 (CD11b^−^CD103^+^) subpopulations from each animal. Sorted DC subsets cells were pelleted and lysed in TRIzol reagent (ThermoFisher, Waltham, MA, USA), then stored at −80 °C.

To assess the capacity of msLN migratory DCs to initiate T_H_2 priming *in vitro*, msLNs were collected on day 8 *p.i.* from *Hp*-infected 1:1 B6:*Cxcr5*^−/−^ chimeras. Tissue was digested with collagenase A and DNase I, then mechanically disrupted as described above. DCs were positively enriched using anti-CD11c microbeads (Miltenyi) in the presence of Fc blocking antibody (anti-CD16/32; 2.4G2 BioXCell), then stained with fluorochrome-conjugated antibodies specific for I-A[b], CD11c, CXCR5, CD11b, CD103, CD45.1, CD45.2, and a lineage dump panel (7AAD, CD3, B220, GR-1, CD64, and NK1.1). Viable migratory CD11c^+^MHCII^hi^Lin^−^ DCs were sort-purified using a BD Aria (in UAB FCSC Core) and further subdivided to obtain CD45.1^+^ (B6) and CD45.2^+^ (*Cxcr5*^−/−^) migratory DCs. To obtain DC subsets from B6 mice and *CD11c-Cxcr5^Tg^* mice, CD11c bead-enriched msLN cells were isolated as described above. Cells were stained with a lineage dump panel (see above) and antibodies specific for CD11c, MHCII, CD11b, CD103, and CXCR5. Viable migratory CD11c^+^MHCII^hi^Lin^−^ cells were sort-purified using a BD Aria (in UAB FCSC Core) and then further subdivided using CXCR5 and the transgene reporter GFP into GFP^+^CXCR5^+^CD11b^+^ (from transgenic mice).

### 2.6. In Vitro T_H_2 Cell Priming Assays

Spleen cells were isolated from naive OT-II.4get mice and CD4 T cells were positively selected using Miltenyi L3T4 beads. DCs were enriched from the msLN of D8 *Hp*-infected mice using anti-CD11c microbeads [Miltenyi] and DC subsets were sort-purified as described above. OT-II.4get transgenic CD4^+^ T cells (2 × 10^5^) were incubated with sorted DC populations (2 × 10^4^) in complete media (RPMI 1640 [Lonza, Walkersville, MD, USA], 10% fetal bovine serum [Biowest, Nuaille, Maine-et-Loire, France], 55 mM 2-mercaptoethanol, 2 mM L-glutamine, 200 mg/mL penicillin, 200 mg/mL streptomycin, 10 mM HEPES, 1 mM sodium pyruvate and 1% MEM nonessential amino acids [all from Corning Mediatech, Glendale, AZ, USA]) in the presence of increasing concentrations of OVA_(323–339)_ peptide (New England Peptide, Gardner, MA, USA) for 4 days. In some experiments recombinant IL-4 (1000 U/mL) and blocking antibody to IFNg (2 mg/mL) were added to the cultures to promote T_H_2-skewing.

In other experiments, recombinant Chi3l1 was tested in the *in vitro* T_H_2 priming assays. Briefly, msLN cells isolated from *Hp*-infected B6 and *Cxcr5*^−/−^ msLNs at 8 days *p.i.* were positively enriched using anti-CD11c beads (Miltenyi) in the presence of anti-CD16/32 (2.4G2 BioXCell). Purified OT-II.4get transgenic CD4^+^ T cells (2 × 10^5^) were incubated with the MACS-enriched DC populations (4 × 10^4^) in complete media in the presence of increasing concentrations of OVA_(323–339)_ peptide (New England Peptide) for 4 days with or without 100 ng recombinant murine CHI3L1 protein (R&D Systems).

OT-II.4get transgenic CD4^+^ T cells were collected on day 4 from the cultures, washed, preincubated with 5 μg/mL anti-CD16/32 to block Fc receptors, and then stained with 7AAD and antibodies for CD4 and CD44. Expression of the IL-4 reporter (GFP) in the live CD4^+^ (7AAD^−^CD4^+^) OT-II cells was detected by flow cytometry using a BD FACSCanto^TM^, (BD Biosciences, San Jose, CA, USA).

### 2.7. DC Chemotaxis Assay

DCs were positively selected with Miltenyi CD11c beads from spleens isolated from uninfected *Cxcr5*^−/−^ and *CD11c*-*Cxcr5^Tg^* mice after mechanical disruption and RBC lysis. Cells were plated (400,000 cells/well) in the upper well of a Costar 24-well transwell plate with a 5 mm pore size polycarbonate filter. DCs were exposed to 500 ng/mL recombinant CXCL13 or media only in the lower well for 90 min, then transmigrated cells were enumerated.

### 2.8. Bulk RNA-Seq Analysis

RNA was isolated (RNeasy micro column (Qiagen, Germantown, MD, USA)) from sorted msLN migratory DCs of *Hp*-infected 1:1 B6:*Cxcr5*^−/−^ mice (described above) with TRIzol (ThermoFisher) followed by Qiagen column clean-up as previously described [35]. Sequencing libraries from three independent experiments which included matched CD45.1^+^ B6 and CD45.2^+^ *Cxcr5*^−/−^ cDC1 (CD103^+^CD11b^−^) and cDC2 (CD103^−^CD11b^+^) cells isolated from the same animals were prepared by GeneWiz LLC (now a subsidiary of Azenta US Inc., Burlington, MA, USA) and sequenced with a 1 × 50 bp single-read (SR) configuration in High Output mode (V4 chemistry) on an Illumina HiSeq2500. Image analysis and base calling were conducted using Hiseq Control Software (HCS). Raw sequencing image files (.bcl files) were converted into fastq files and de-multiplexed using Illumina’s bcl2fastq 2.17 software. One mismatch was allowed for index sequence identification. Adapter content was removed from fastq files using Skewer 0.2.2 [36] and data aligned with STAR 2.5.3a [37] to the ENSEMBLE C57BL/6J GCA_000001635.9 reference mouse genome and transcriptome. PCR duplicate reads were flagged using PICARD MarkDupilicates 1.127 (http://broadinstitute.github.io/picard (accessed on 17 January 2024), gene counts for each sample were computed using GenomicRanges 1.34.0 [38], and reads per kilobase per million (RPKM) normalized in R 3.5.2. 11,625 expressed genes (Appendix A) were identified with expressed genes defined as genes with at least 3 reads per million (RPM) in all samples derived from at least one DC subset (i.e., WT and/or *Cxcr5*^−/−^ cDC2 and/or cDC1 cells). The differential expression of gene (DEG) analysis presented in Figure 4 was restricted to expressed genes with at least 1 RPKM average in WT or *Cxcr5*^−/−^ cDC2 or cDC1 cells, which yielded 10,774 expressed genes in cDC2 and 10,644 expressed genes in cDC1 cells. DEGs between groups were identified using DESeq2 1.26.0 [39]. Using an FDR cutoff of q < 0.05 and log_2_FC of >1 or <−1, 51 DEGs were identified in the comparison between WT and *Cxcr5*^−/−^ cDC2 cells and 25 DEGs were identified in the comparison of WT and *Cxcr5*^−/−^ cDC1 cells (Appendix A). Gene set enrichment analysis (GSEA, [40]) was performed (Appendix A) using the HALLMARK gene set database and the GSEA PreRanked analysis program (http://software.broadinstitute.org/gsea/index.jsp (accessed on 17 January 2024)). The ranked list of expressed genes from WT and *Cxcr5*^−/−^ DCs was determined by multiplying the −log_10_ of the *p*-value from DESeq2 by the sign of the fold change. The 85 genes from cDC2 cells that met a threshold of FDR q < 0.05 (comparison of B6 cDC2 cells over *Cxcr5*^−/−^ cDC2 cells, see Appendix A) were submitted to Ingenuity Pathway Analysis (IPA, QIAGEN Digital Insights) to identify significantly enriched pathways (Appendix A) and upstream regulators (Appendix A). RNA-seq data sets were deposited in the NCBI Gene Expression Omnibus (GSE301250).

### 2.9. Statistical Analysis

Statistical details of all experiments including tests used, n, and number of experimental repeats are provided in figure legends. FlowJo (version 9, Tree Star) was used for flow cytometric analyses. Prism Graphpad (version 10.1) was used for statistical analyses and graphing except where indicated. Details of transcriptomics statistical analyses are provided above.

## 3. Results

### 3.1. T_H_2 Priming In Vitro by Hp-Induced DCs Requires DC Intrinsic Expression of CXCR5

Our prior *in vivo* studies revealed that the T_H_2 response to *H. polygyrus* (*Hp*) requires CXCR5-expressing DCs [3]. At that time, we postulated that CXCR5 primarily serves to facilitate the recruitment and placement of the DCs at the border of the B cell follicle within the mesenteric lymph node (msLN), thereby allowing the CXCR5^+^ CD4 T cells to interact with both the DCs and B cells that collectively support T_H_2 and T_FH_ commitment [3,25]. However, it was also possible that CXCR5 expression by the DCs resulted in placement of the DCs in a microenvironment that programmed the T_H_2 priming capacity of the CXCR5^+^ DCs. If so, we predicted that *Cxcr5* deficient (*Cxcr5*^−/−^) DCs would be unable to induce T_H_2 development even when both T cells and DCs were co-localized in the same *in vitro* cultures. To test this hypothesis, we set up a T_H_2 priming *in vitro* culture assay using naïve CD4 T cells and different populations of DCs. Since some LNs are missing in mice with a global deletion of *Cxcr5* (*Cxcr5*^−/−^) and the remaining LNs exhibit disorganized architecture [29], we first generated mixed bone marrow (BM) chimeras by reconstituting lethally irradiated wildtype (WT) C57BL6/J CD45.1^+^ congenic mice (B6 CD45.1^+^) with a 1:1 ratio of B6 CD45.1^+^ BM and B6.*Cxcr5*^−/−^ (CD45.2^+^) BM (1:1 B6:*Cxcr5*^−/−^ chimeras, Figure 1A). Following 8 weeks of reconstitution, we infected the chimeric mice by gavage with 200 *Hp* L3 larvae. On day 8 post-infection (D8 *p.i.*) we sort-purified CD45.1^+^ WT and CD45.2^+^ *Cxcr5*^−/−^ migratory CD11c^+^MHCII^hi^ DCs (Figure 1B) isolated from the msLN of the infected mice. We cultured the purified migratory DCs in the presence of increasing concentrations of OVA peptide (OVAp) and purified splenic OVA-specific CD4^+^ TCR transgenic OT-II cells that can express an IL-4 mRNA reporter gene, GFP [41] (OTII.4get T cells, Figure 1C). On day 4, we measured GFP expression by the OVA-specific OTII.4get T cells in the cultures. We observed that the frequency of IL-4-expressing GFP^+^ T cells was significantly reduced when T cells were co-cultured with *Cxcr5*^−/−^ migratory DCs and low (0.01 mM OVAp) concentrations of OVAp (T_H_0 conditions, Figure 1D,E). However, this peptide dose was sufficient to support WT DC priming of T_H_2 cells (Figure 1D,E). When recombinant IL-4 and blocking antibodies to IFNg were added to the cultures to enforce skewing of the naïve T cells to the T_H_2 fate (T_H_2 conditions), we observed a large proportion (40–75%) of the OTII.4get T cells expressed IL-4/GFP^+^ and we found no differences between the cultures containing WT or *Cxcr5*^−/−^ migratory DCs (Figure 1F). These data therefore suggest that intrinsic expression of CXCR5 by an antigen-bearing migratory DC population is necessary for efficient T_H_2 priming, particularly when antigen availability is low and polarizing cytokines are not present.

### 3.2. CXCR5 Is Expressed by Migratory cDC2 and Not cDC1 Cells

Our results indicated that *Cxcr5* expression by migratory DCs was not only required for their recruitment to the perifollicular region of the msLN following *Hp* infection [3] but appeared important for these DCs to acquire T_H_2 priming capacity. Migratory CD11c^+^MHCII^hi^ DCs can be subdivided into cDC1 cells (CD103^+^CD11b^−^), cDC2 cells (CD103^−^CD11b^+^), as well as the double positive (DP, CD103^+^CD11b^+^) and double negative (DN, CD103^−^CD11b^−^) populations that are believed to be part of the broader cDC2 compartment [7,8,9,10]. The CD11b^+^ migratory DC population, which includes both cDC2 and DP cells, was previously shown to initiate T_H_2 development *in vitro* [19] and to express CXCR5 after injection with LPS and OVA nanoparticles [21]. Therefore, we hypothesized that the msLN migratory CD11b^+^CD103^−^ cDC2 and/or DP DCs were likely to express CXCR5 following *Hp* infection. To test this, we infected B6 mice with *Hp* and on D8 *p.i.* we analyzed CXCR5 expression by the msLN cells. As expected, CXCR5, a canonical chemokine receptor expressed by B cells, was highly expressed by essentially all B220^+^ B cells (Figure 2A,B). Next, we examined CXCR5 expression by the migratory DC subpopulations (Figure 2C) found in the msLN of *Hp*-infected mice. Following magnetic bead purification to enrich for the msLN CD11c-expressing cells, we observed that cDC1 cells did not express detectable levels of cell surface CXCR5 (Figure 2D,E). By contrast, some DCs within each of other subsets did express CXCR5 (Figure 2D,F–H). The CD11b^+^CD103^−^ cDC2 cells, which were largely CXCR5^+^ (Figure 2H), also expressed the highest levels of CXCR5, albeit at low levels relative to the B cells. Given the known role for the cDC2 cells in directing T_H_2 development to nematodes [14,18,19,20], we postulated that CXCR5 expression by migratory cDC2 cells may regulate the T_H_2 priming capability of these cells.

### 3.3. Enforced CXCR5 Expression by Migratory CD11b^+^ DCs Enhances T_H_2 Priming In Vitro

Since CXCR5 expression by migratory DCs regulated the capacity of these cells to induce T_H_2 development *in vitro*, we postulated that enforced expression of CXCR5 by DCs that did not normally express CXCR5, like the CD11b^neg^CD103^+^ cDC1 population, might allow these cells to acquire T_H_2 priming capabilities. To address this, we examined mice that were engineered to express a CXCR5 transgene specifically by CD11c^+^ cells. Briefly, we generated mice in which a loxP-STOP-loxP-Cxcr5-IRES-EGP construct was knocked into the ROSA26 locus (Appendix A) and then intercrossed the *Cxcr5^Tg^* knock-in mice with CD11c-cre transgenic mice. In these mice (*CD11c-Cxcr5^Tg^*, Appendix A), expression of Cre by the CD11c^+^ cells results in removal of the STOP codons upstream of *Cxcr5*, which allows for ROSA26-directed transcription of CXCR5 and the EGFP reporter in the CD11c-expressing cells. Consistent with prior reports showing that CD11c is expressed on a small subset of B cells [42], expression of the GFP reporter was restricted to a minor fraction of the CD19^+^ B cells (Figure 3A). By contrast, many, but not all, CD11c^+^ DCs from the msLNs of *CD11c-Cxcr5^Tg^* mice expressed the reporter gene, EGFP (Figure 3B). These GFP^+^ DCs also co-expressed CXCR5 (Figure 3B). To test whether the CXCR5 transgene was functional in the CD11c DCs, we purified splenic GFP-expressing DCs and assessed the chemotaxis of these cells to the CXCR5 ligand, CXCL13. Chemotaxis to CXCL13 was easily measured in the wells containing GFP^+^ CXCR5-expressing splenic DCs (Figure 3C) but was not detected in wells containing purified *Cxcr5*^−/−^ splenic DCs isolated from *Cxcr5*^−/−^ mice (Figure 3C). Thus, the CXCR5 transgene was expressed and functional in at least a subset of the DCs isolated from *CD11c-Cxcr5^Tg^* mice.

Although we anticipated that all the CD11c^+^ DCs should express the CXCR5 transgene and GFP reporter, there were clearly DCs that did not express either the reporter or CXCR5 (Figure 3B). To assess whether the transgene expression was limited to specific populations of DCs, we determined the frequency of GFP expression by different DC subsets before and after *Hp* infection of the *CD11c-Cxcr5^Tg^* mice. We found that within the resident DC subset (CD11c^+^MHCII^int^Lin^neg^, Appendix A) present in the msLNs of *CD11c-Cxcr5^Tg^* mice, 32% expressed GFP constitutively (Appendix A) and that the frequency (Appendix A) and number (Figure 3D,E) of these GFP-expressing msLN resident DCs declined modestly following *Hp* infection. When we examined the migratory (CD11c^+^MHCII^hi^Lin^neg^) DC subsets in uninfected animals, we found that ~30% of the cDC2 cells and DP cells from *CD11c-Cxcr5*^Tg^ mice expressed the GFP reporter (Appendix A) and the frequency (Appendix A) and number (Figure 3D,E) of these GFP^+^ cDC2 and DP cells modestly increased following infection. By contrast, very few (3–6%) of the cDC1 and DN cells expressed the CXCR5 reporter either before or after infection (Appendix A) and the numbers of CXCR5 expressing DN and cDC1 cells was very low (Figure 3D,E). Thus, enforced expression of CXCR5 under the control of the ubiquitous promoter ROSA26 was observed in many but not all migratory DCs linked to the cDC2 compartment but was not seen in the migratory cDC1 cells.

Since the CXCR5 transgene could not be detected in the CD103^+^CD11b^−^ cDC1 cells, we were unable to test whether expression of CXCR5 by cDC1 cells was sufficient to endow these cells with T_H_2 priming capability. However, we noted that the EGFP^+^ cDC2 cells from the *CD11c-Cxcr5*^Tg^ mice expressed approximately 10-fold higher levels of cell surface CXCR5 (Figure 3F,G) relative to cDC2 cells from the non-transgenic B6 mice (see Figure 2D). To address whether increased expression of CXCR5 by CD11b^+^ migratory DCs correlated with enhanced T_H_2 priming capacity *in vitro*, we measured IL-4 expression by OT-II.4get T cells that were co-cultured for 4 days with 0.01 mM OVAp in the presence of GFP^+^ CD11b^+^ migratory DCs (which includes both cDC2 and DP cells, see Appendix A) purified from msLNs of D8 *Hp*-infected *CD11c-Cxcr5*^Tg^ mice or with CD11b^+^ migratory msLN DCs from infected B6 mice. We observed that the frequency of IL-4/GFP^+^ T cells almost doubled when the T cells were co-cultured with GFP^+^ DCs from the *CD11c-Cxcr5*^Tg^ mice (Figure 3H,I). Therefore, overexpression of CXCR5 by CD11b^+^ migratory DCs enhances the T_H_2 priming capacity of these DCs while deleting *Cxcr5* from migratory DCs attenuates their T_H_2 priming capacity.

### 3.4. CXCR5 Regulates Cell Division Pathways and Modulates the Size of the Migratory cDC2 Compartment

Although we previously speculated that CXCR5 expressed by migratory DCs simply ensured co-localization of these DCs with the CXCR5-expressing CD4 T cells that are endowed with the potential to differentiate into T_H_2 and T_FH_ cells [3], our *in vitro* T_H_2 priming experiments suggested that DCs lacking CXCR5 were not able to receive appropriate T_H_2 programming instructions. To test this possibility, we performed bulk RNA-seq analysis on sort-purified CD45.1^+^ WT and CD45.2^+^ *Cxcr5*^−/−^ CD11b^neg^CD103^+^ cDC1 cells and CD11b^+^CD103^neg^ cDC2 cells isolated from msLNs of D8 *Hp*-infected 1:1 B6:*Cxcr5*^−/−^ chimeric mice. Migratory cDC1 and cDC2 sorting strategies are shown in Appendix A. Overall, this analysis only identified a modest number of differentially expressed genes (DEG; FDR *p* < 0.05 and log_2_FC ≥ ±1) (Appendix A), which included 25 DEGs when comparing WT and *Cxcr5*^−/−^ cDC1 cells (Figure 4A) and 51 DEGs when comparing WT and *Cxcr5*^−/−^ cDC2 cells (Figure 4B). Greater than >90% of the DEGs between WT and *Cxcr5*^−/−^ cDC2 cells were upregulated in WT cDC2 cells (Figure 4B), suggesting that the CXCR5^+^ cDC2 cells might receive environment-specific activation signals when localized within the perifollicular region of the LN. We next performed gene set enrichment analysis (GSEA) comparing the Hallmark gene sets [40] to the ranked list of genes expressed in WT over *Cxcr5*^−/−^ cDC2 cells. GSEA (Appendix A) revealed significant enrichment in hallmark gene sets for E2F (Figure 4C) and G2M checkpoint (Figure 4D) targets in the WT cDC2 cells relative to *Cxcr5*^−/−^ cDC2 cells. To confirm these findings, we performed Ingenuity Pathway Analysis (IPA) on the 85 genes (Appendix A) that met an FDR q < 0.05 threshold when comparing expression between WT and *Cxcr5*^−/−^ cDC2 cells. Again, we found that signaling pathways related to cell cycle checkpoint and mitosis were predicted to be activated in WT cDC2 cells relative to *Cxcr5*^−/−^ cDC2 cells (Figure 4E, Appendix A). Consistent with these findings, upstream regulators associated with proliferation or survival including the c-myc regulating transcription factor Zbtb17 (Miz-1 [43]) and GM-CSF receptor (Csf2 [44]), were predicted to be activated in B6 cDC2 cells relative to the paired *Cxcr5*^−/−^ cDC2 cells isolated from the same animals (Figure 4F, Appendix A). Moreover, cell division associated genes like *Ccna2, Ccnb2*, *Top2a, Kif20a* and *Nuf2* were each upregulated 1.9–2.1 fold in WT compared to *Cxcr5*^−/−^ cDC2 cells and were identified as common target molecules in the IPA analyses comparing WT and *Cxcr5*^−/−^ cDC2 cells (Appendix A). These data suggested that WT and *Cxcr5*^−/−^ cDC2 cells may differ in their proliferative potential.

To address whether CXCR5 expression by cDC2 cells influenced the size of the cDC2 population in the msLN during *Hp* infection, we determined the number of migratory DCs present in the msLNs from D8 *Hp*-infected 1:1 B6:*Cxcr5*^−/−^ chimeric mice. As expected, the ratio of CD45.1^+^ and CD45.2^+^ cells in the msLN was roughly 1:1 and mirrored the input of transferred BM (Appendix A), suggesting no significant differences in recovery of CD45^+^ leukocytes of WT or *Cxcr5*^−/−^ origin. Consistent with this, when we examined the migratory DC subsets (Appendix A) in each chimeric animal, we found that the numbers of WT and *Cxcr5*^−/−^ cDC1 (Figure 4G) and DP DCs (Figure 4H) were not significantly different when analyzed with a paired *t* test. However, the numbers of CD45.2^+^ *Cxcr5*^−/−^ DN (Figure 4I) and cDC2 (Figure 4J) cells were significantly decreased when compared to the paired CD45.1^+^WT DN DCs or cDC2 cells from the same animals. Similar results were observed when we calculated the frequencies of the WT and *Cxcr5*^−/−^ DC subsets present in the same mice (Appendix A). Thus, the loss of CXCR5 on migratory cDC2 cells appeared alter the size of the *Hp*-elicited cDC2 and DN response in the msLN. Given that the WT and *Cxcr5*^−/−^ cells were collected from the same host msLN and that the transcriptional pathways regulating cell division were significantly inhibited in the *Cxcr5*^−/−^ migratory cDC2 cells, we conclude that CXCR5 likely regulates both the recruitment and homeostasis of these migratory DC populations within the LN following infection.

### 3.5. Chi3l1, a CXCR5-Regulated Gene in cDC2 Cells, Supports T_H_2 Priming In Vitro

Our data showed that *Cxcr5* deficiency in DCs resulted in decreased numbers of migratory cDC2 cells in the msLN of *Hp*-infected mice (Figure 4J), suggesting that CXCR5 expression is important for the expansion and/or maintenance of this subset in the msLN following *Hp* infection. However, our data also indicated an intrinsic defect in the capacity of *Cxcr5*^−/−^ DCs to support T_H_2 priming *in vitro* (Figure 1E). We therefore examined whether other CXCR5-dependent genes might influence the T_H_2 priming ability of the cDC2 cells. The *Chi3l1* (chitinase 3-like-1) gene, which encodes a secreted protein (Chi3l1 in mice or YKL40 in humans) and is associated with allergic responses in both humans and mice [26], was the most upregulated gene (log_2_ FC 7.1 and FDR 4.3 × 10^−48^) in WT relative to *Cxcr5*^−/−^ cDC2 cells (Figure 4B). In fact, *Chi3l1* was essentially absent in *Cxcr5*^−/−^ cDC2 cells and was not expressed by either WT or *Cxcr5*^−/−^ cDC1 cells (Figure 5A). This indicated that *Chi3l1* was selectively expressed by cDC2 cells, and its expression was controlled by CXCR5.

To determine whether soluble Chi3l1 protein could enhance T_H_2 priming by cDC2 cells, we co-cultured purified naïve OT-II.4get CD4 T cells with low dose OVAp and purified CD11b^+^ migratory DCs (cDC2 and DP cells) from the msLN of D8 *Hp*-infected WT mice DCs in the presence and absence of recombinant Chi3l1. On D4, we determined the frequency of T cells expressing the IL-4 reporter GFP and found that inclusion of Chi3L1 protein in the cultures significantly enhanced the development of T_H_2 cells *in vitro* (Figure 5B,C). Next, we asked whether we could rescue the T_H_2 priming capacity of the *Cxcr5*^−/−^ migratory DCs by providing Chi3l1 protein *in trans*. However, addition of exogenous Chi3l1 to *in vitro* cultures containing OT-II.4get CD4 T cells with low dose OVAp and purified CD11b^+^ migratory WT or *Cxcr5*^−/−^ DCs did not rescue T_H_2 priming by the *Cxcr5*^−/−^ DCs (Figure 5D). Therefore Chi3l1, which is expressed in a CXCR5-dependent fashion by cDC2 from the msLNs of *Hp*-infected mice, can enhance T_H_2 induction *in vitro*. However, Chi3l1 protein is not sufficient on its own to repair the T_H_2 priming capacity of *Cxcr5*^−/−^ migratory DCs *in vitro*, suggesting that CXCR5 plays a multi-factorial role in promoting T_H_2 development.

### 3.6. Chi3l1 Regulates the Size of the msLN cDC2 Population Following Hp Infection

Given that *Chi3l1* is expressed in a *Cxcr5*-dependent fashion by msLN cDC2 on D8 post-*Hp* infection and can support T_H_2 priming *in vitro*, we postulated that this factor might regulate the cDC2 population that drives T_H_2 responses *in vivo*. To test this, we compared the migratory DC populations present in the msLN of uninfected and *Hp*-infected WT and *Chi3l1*^−/−^ mice. We found that the number of total msLN migratory CD11c^+^MHCII^hi^ DCs was the same in uninfected WT and *Chi3l1*^−/−^ mice and was modestly reduced in the msLN *Chi3l1*^−/−^ mice on D8 post-*Hp* infection (Appendix A). Analysis of the four migratory DC subsets (Appendix A) following infection revealed that the number of migratory CD11b^+^CD103^−^ cDC2 (Figure 5E) and DN (Figure 5F) cells was significantly decreased in the msLN of *Hp*-infected *Chi3l1*^−/−^ mice relative to WT mice. By contrast, no reduction in the number of cDC1 or DP migratory DCs was observed in the *Chi3l1*^−/−^ msLN (Figure 5G,H). Similar results were observed when we examined the frequencies of the different msLN DC subsets in the *Hp*-infected WT and *Chi3l1*^−/−^ mice (Appendix A). Thus, deletion of either *Cxcr5* or *Chi3l1* decreases the number of migratory cDC2 cells in the msLNs of *Hp*-infected mice.

### 3.7. DC-Intrinsic Expression of Chi3l1 Regulates T_H_2 Responses In Vivo

Prior experiments from our group showed that the number of IL-4^+^IL-13^+^ producing T_H_2 cells was significantly decreased in the msLN of *Hp*-infected *Chi3l1*^−/−^ mice and that *Chi3l1* expression by hematopoietic cells was required for development of IL4^+^IL-13^+^ T_H_2 cells [45]. We also observed that optimal IL-4 production by restimulated CD4 T cells isolated from the msLN of *Hp*-infected mice required T cell intrinsic expression of *Chi3l1* [45]. Given our results showing that *Chi3l1* is expressed by the cDC2 migratory DCs, which can support T_H_2 responses *in vitro* and *in vivo* [14,18,19,20], we evaluated whether *Chi3l1* deficiency selectively in DCs would impact T_H_2 priming following *Hp* infection. To test this, we reconstituted irradiated wildtype BALB/c mice with a 5:1 ratio of CD11c-DTR Thy1.1^+^ BM and either wildtype BALB/c (Thy1.1^neg^) BM or Balb/c *Chi3l1*^−/−^ (Thy1.1^neg^) BM (Figure 6A). At 8 weeks post-reconstitution, we infected the chimeric mice with *Hp* and injected the mice with diphtheria toxin (DT) to deplete the wildtype DCs derived from the CD11c-DTR BM cells. This left the mice with DCs derived from either wildtype (DTR-WT) or *Chi3l1*^−/−^ (DTR-*Chi3l1*^−/−^) BM to prime the CD4 T cell response to *Hp* (Figure 6A). Since all T cells derived from the Thy1.1^+^CD11c-DTR BM were wildtype and competent to express *Chi3l1* (Figure 6A), these chimeras could be used to analyze the expansion and cytokine production of the wildtype Thy1.1^+^ CD4 T cells in the presence of either wildtype or *Chi3l1*^−/−^ DCs. Using this model, we observed decreased numbers of wildtype Thy1.1^+^ msLN CD4 T cells in DTR-*Chi3l1*^−/−^ chimeras compared to DTR-WT chimeras (Figure 6B). This suggested that *Chi3l1* expression by DCs was important for CD4 T cell accumulation in the msLN post-*Hp* infection. Moreover, we observed that fewer CD4^+^CD44^hi^Thy1.1^+^ CD4 T cells from the DTR-*Chi3l1*^−/−^ msLNs produced IL-13 and IL-4 (Figure 6C,D) following *in vitro* anti-CD3 restimulation. Thus, *Chi3l1* expression specifically by DCs supports optimal T_H_2 development following *Hp* infection.

These data show that expression of CXCR5 by migratory CD11b^+^ DCs is required to initiate T_H_2 priming, particularly when antigen availability is limited and T_H_2 polarizing cytokines are not abundant. CXCR5 expression by the migratory cDC2 cells supports (i) their trafficking to the LN perifollicular region where the DCs can interact with CXCR5-expressing activated CD4 T cells and (ii) places these DCs within a niche where the DCs appear to receive additional conditioning signals. These location-provided cues control the size of the migratory cDC2 population and instruct these cells to produce factors, like Chi3l1, that further support T_H_2 priming and commitment. We discuss how adjuvants engaging this pathway might enhance protective T_H_2 immunity to parasitic infections while drugs that block the activity of Chi3l1 in DCs and CD4 T cells might prevent damaging T_H_2 responses in the settings of allergic inflammation.

## 4. Discussion

More than a decade ago, we reported that CXCR5 expression by DCs promotes T_H_2 differentiation following *Hp* infection [3]. Since that time, we have come to appreciate that there are multiple lineages of migratory DCs, which require distinct transcription factors for development and are endowed with different functional attributes [1,2]. It is also known that migratory DC subsets express different arrays of chemokine receptors, which allow the cells to localize in distinct areas of the LN, with CCR7^+^ IL-12 expressing migratory DCs occupying the paracortex and migratory CXCR5^+^ DCs residing outside of the T cell zone in close proximity to the B cell follicles in the inter- or peri-follicular area [21,25,46]. Consistent with this, we found that *Il12a* (p35), *Il12b* (p40) and *Ccr7* were more highly expressed by msLN cDC1 cells relative to msLN cDC2 cells from *Hp*-infected B6 mice (see Appendix A). By contrast, we found that CXCR5 was expressed by cells in the migratory cDC2 compartment and was not expressed by cDC1 cells. Given the literature showing an important role for cDC2 cells in T_H_2 priming [5,7,11,12], our data argue that CXCR5 expressed specifically by cells in the cDC2 compartment is important for T_H_2 development. However, it was unclear whether CXCR5 expressed by cDC2 cells was needed to co-localize the DCs in the same region of the LN as the CD4 T cells that go on to differentiate into T_H_2 cells, whether the CXCR5-expressing cDC2 cells receive programming signals within the perifollicular region that enabled the DCs to initiate T_H_2 development, whether signaling through CXCR5 suppressed CXCR5^+^ cDC2 cells from expressing the T_H_1-inducing cytokine IL-12, or some combination of these possibilities. We found that deletion of *Cxcr5* in cDC2 cells did not result in upregulation of *Il12a* or *Il12b*, which suggested that CXCR5 expression by the DCs was not directly or indirectly preventing these DCs from making IL-12. Rather, CXCR5 seemed to support the acquisition of T_H_2 priming capabilities by the cDC2 cells as *Cxcr5*^−/−^ migratory DCs less effectively initiated T_H_2 development even *in vitro* when the DCs and naïve T cells were co-cultured together with exogenously provided peptide. Moreover, overexpression of CXCR5 within the CD11b^+^ migratory DCs further enhanced the *in vitro* T_H_2 priming capacity of cDC2 cells. Taken together, these data suggest that CXCR5 expression by cDC2 cells is not only important for co-localization of cDC2 cells and CD4 T cells outside the T cell zone but also enables the DCs to acquire optimal T_H_2 priming capability.

To better understand what pathways or molecules enhance the T_H_2-promoting capacity of the CXCR5^+^ migratory DCs, we analyzed the transcriptome of B6 wildtype and *Cxcr5*^−/−^ cDC2 cells isolated from the msLN of *Hp*-infected 1:1 B6:*Cxcr5*^−/−^ chimeric mice. In this analysis, the wildtype and *Cxcr5*^−/−^ cells developed in the same wildtype hosts, which allowed us to address the DC intrinsic role for CXCR5 in regulating T_H_2 responses to *Hp* infection in animals that expressed CXCR5 during embryogenesis and underwent normal LN development [47]. While the number of genes that differed in expression between B6 and *Cxcr5*^−/−^ cDC2 cells was relatively small, one gene in particular, *Chi3l1*, stood out as it was the most differentially regulated gene between the B6 and *Cxcr5*^−/−^ cDC2 cells and was not expressed by cDC1 cells. Published human studies show that expression of the *CHI3L1* gene and YKL-40 protein (the protein encoded by *CHI3L1*) is associated with food allergy, allergic rhinitis, and atopic dermatitis [48,49,50,51,52,53,54]. Similarly, analyses of *Chi3l1*^−/−^ mice and Chi3l1 blocking antibodies in allergic mouse models [26,50,52,54,55,56,57] and our own prior work evaluating *Chi3l1*^−/−^ mice infected with *Hp* [45] reveal that *Chi3l1* serves as a key regulator of T_H_2 immunity. In our earlier study, we used reciprocal BM chimeras to demonstrate that intrinsic expression of *Chi3l1* within the T cell compartment is necessary for T_H_2 development following *Hp* infection. However, the data also suggested that *Chi3l1* made by other cell types might also be important for optimal T_H_2 development following nematode infection. Although Chi3l1 is a secreted protein, it expresses a nuclear localization signal that permits its re-importation to the nucleus, as reported for human monocyte-derived DCs cultured *in vitro* [58]. Thus, Chi3l1 has the potential to act via a paracrine mechanism on surrounding cells or via autocrine mechanisms to intrinsically influence gene expression in the cell that expresses Chi3l1. Our new data showed that direct administration of soluble Chi3l1 protein to co-cultures of DC and CD4 T cells enhanced T_H_2 development *in vitro*, suggesting that secreted Chi3l1 can act on the CD4 T cells and/or the DCs to promote T_H_2 development. However, we also showed that deletion of *Chi3l1* specifically in the DCs was sufficient to impair the T_H_2 response *in vivo*, even when the T cells were wild type and competent to express the *Chi3l1* gene. Collectively, these data strongly suggest that Chi3l1 expression by both CD4 T cells and the CXCR5^+^ cDC2 cells promotes T_H_2 development following nematode infection.

Upregulation of *Chi3l1* by *Hp*-infected migratory cDC2 cells required CXCR5 expression by the cDC2 cells. However, addition of recombinant Chi3l1 to co-cultures containing *Cxcr5*^−/−^ migratory DCs and CD4 T cells was not sufficient to rescue T_H_2 priming *in vitro*. These data therefore indicated that CXCR5 expression must promote the T_H_2 priming capacity of migratory cDC2 cells in a multi-factorial manner. While we cannot say with certainty what other conditioning signals are provided to the migratory DCs that reside outside of the T cell zone, we know that deletion of either *Cxcr5* or *Chi3l1* in DCs results in altered homeostasis of the cDC2 compartment with reductions in both the CD11b^+^CD103^−^ cDC2 cells and the CD11b^−^CD103^−^ DN cells, which, at least in skin, are thought to be part of the cDC2 lineage [7,8,9,10]. This altered DC homeostasis was particularly notable in the 1:1 B6:*Cxcr5*^−/−^ chimeric mice when the *Cxcr5*^−/−^ DCs and their precursors were required to directly compete with the B6 wildtype DCs in the same host. When we used GSEA and IPA to compare the transcriptional profile of the B6 wildtype and *Cxcr5*^−/−^ CD11b^+^CD103^−^ cDC2 cells from the 1:1 B6:*Cxcr5*^−/−^ chimeric mice, we observed enrichment of gene signatures and pathways associated with cell division, cell cycle, G2/M checkpoint, and mitotic spindle in the wildtype cDC2 cells relative to the *Cxcr5*^−/−^ cDC2 cells. Predicted upstream regulators of the B6 wildtype cDC2 transcriptome included Zbtb17 and PGE2, which are both implicated in suppression of DC *Il12* expression [59,60,61,62], and CSF2 (GM-CSF), which in some models increases DC T_H_2 priming [44,63,64]. Consistent with a more pro-proliferative signature, Zbtb17 regulates cell cycle via interaction with Myc [65,66] while prostaglandin receptor E2 (PGE2) and CSF2 (GM-CSF), support the activation and proliferation of myeloid-lineage cells [60,67,68,69]. These results were initially somewhat surprising as migratory DCs have been historically viewed as terminally differentiated cells that do not proliferate and have a relatively short lifespan [2]. However, a recently published analysis of macrophage populations reveals that gene expression patterns of cell cycle genes vary between classically activated macrophages induced by interferon gamma and alternatively activated macrophages induced by IL-4 [70]. Both populations express cell cycle-sensitive transcriptional profiles, but the canonical genes associated with classically activated macrophages are generally upregulated in the non-proliferating (in G1 phase) classically activated macrophages while genes associated with polarization to the T_H_2 and tissue remodeling programs of the alternatively activated macrophages are more often upregulated in proliferating (G2/M phase) alternatively activated macrophages [70]. These results suggest that genes regulating expansion/maintenance and functional attributes of macrophages are controlled in a cell cycle dependent manner. A similar regulatory mechanism has not been described in migratory DCs from helminth-infected mice, but it is tempting to speculate that the CXCR5 expression by the cDC2 population might position the DCs in a niche within the msLN that favors continued proliferation and the acquisition of optimal T_H_2 priming capabilities.

## 5. Conclusions

Taken together, the data presented here support the conclusion that the CXCR5 is expressed by migratory DCs within the cDC2 compartment that responds to intestinal helminth infection by supporting development of the protective T_H_2 response. CXCR5 expression by these DCs is necessary for optimal T_H_2 priming both *in vitro* and *in vivo* and supports the continued maintenance and proliferative signature of the cDC2 cells. CXCR5 also facilitates cDC2 cell dependent upregulation of *Chi3l1*—a gene that encodes a secretory protein that can enhance T_H_2 development both *in vitro* and *in vivo*. Thus, CXCR5 expression by the DCs does more than simply facilitating co-localization of the antigen-presenting DCs and CD4 T cells outside of the T cell zone, as CXCR5 also appears to “park” the DCs within a niche that supports their maintenance, proliferation and functional attributes. We expect that this more nuanced understanding of how CXCR5^+^ cDC2 cells support T_H_2 responses to intestinal parasites like *Hp* may, in the future, inform the design of anti-helminth vaccines. In particular, we believe that Chi3l1, which is associated with T_H_2 responses in both mice and humans, is an interesting target. In the setting of vaccination, we propose that adjuvants that are designed to specifically elicit development and expansion of the CXCR5^+^ cDC2 compartment and induce expression of Chi3l1 by these DCs may be particularly effective activators of protective T_H_2 and T_FH_ immunity.

## Figures and Tables

**Figure 1 vaccines-13-00943-f001:**
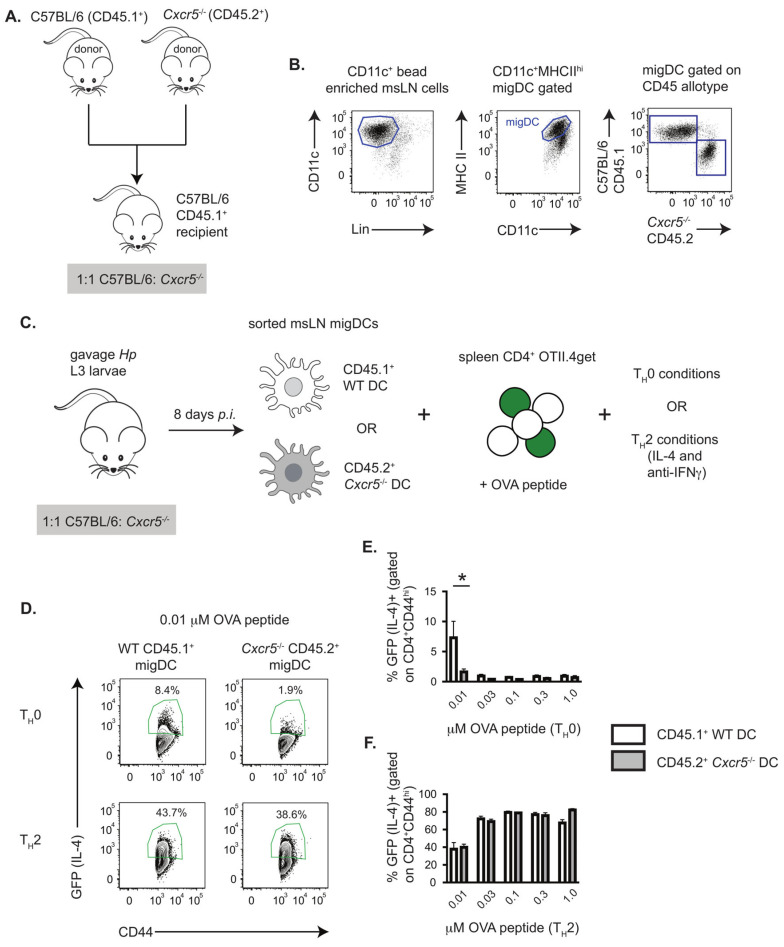
CXCR5 expression by mLN migratory DCs from *Hp*-infected mice promotes T_H_2 priming *in vitro*. (**A**–**C**) Description of *in vitro* T_H_2 priming cultures using msLN migratory DCs from *Hp*-infected mice and CD4 T cells from OTII-4get IL4 reporter mice. CD45.1^+^ B6 recipient mice (n = 15) were irradiated and reconstituted with a 1:1 ratio of wildtype B6.CD45.1^+^ and *Cxcr5*^−/−^ CD45.2^+^ BM cells, as shown in (**A**), to generate 1:1 B6:*Cxcr5*^−/−^ chimeric mice. Chimeric mice were infected 8 weeks post-reconstitution with 200 *Hp* stage L3 larvae and msLN cells were isolated on day 8 post-infection. DCs were enriched with CD11c MACS beads and wildtype (CD45.1^+^) or *Cxcr5*^−/−^ (CD45.2^+^) CD11c^+^MHCII^hi^ migratory DCs (**B**) were sort-purified and then incubated at a 1:10 ratio with splenic CD4 T cells (**C**) isolated from uninfected OTII.4get mice. Cultures included increasing concentrations of OVA peptide in the absence (T_H_0 conditions) or presence (T_H_2 conditions) of recombinant IL-4 (1000 U/mL) and blocking antibody to IFNg (2 mg/mL). (**D**–**F**) CD4 T cells (7AAD^−^CD4^+^CD44^hi^) from the DC/OTII.4get T cell co-cultures were analyzed for expression of the IL-4 reporter, GFP, on day 4. Representative flow cytometry plots (**D**) showing GFP expression by OTII.4get CD4 T cells. The frequencies of IL-4 reporter expressing OTII.4get CD4 T cells in triplicate cultures containing CD45.1^+^ WT or CD45.2^+^ *Cxcr5*^−/−^ DCs under T_H_0 (**E**) and T_H_2 (**F**) conditions are reported. Data representative of ≥3 independent experiments. Statistical analysis performed with unpaired 2-tailed Student’s *t*-test. * *p* ≤ 0.05. Portions of panel (**C**) created in BioRender. Curtiss, M. (2025) https://BioRender.com/ebwsrlb (accessed on 28 August 2025).

**Figure 2 vaccines-13-00943-f002:**
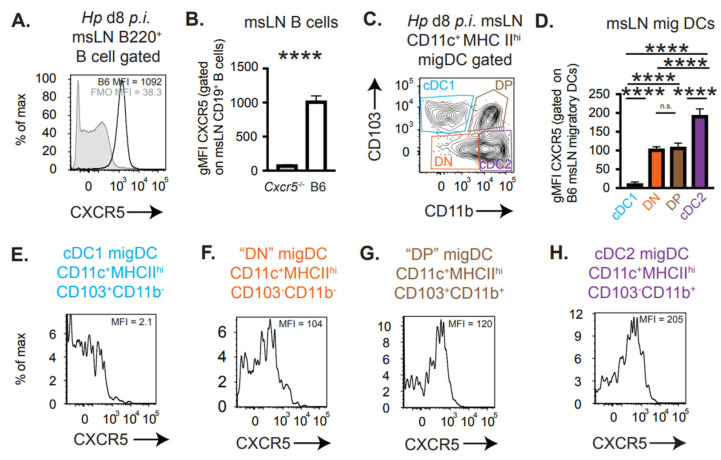
CXCR5 expression by msLN migratory DCs is restricted to the cDC2 compartment following *Hp* infection. (**A**,**B**) CXCR5 expression by msLN B cells from d8 *Hp*-infected B6 (n = 5) and *Cxcr5*^−/−^ (n = 3) mice. Representative histogram (**A**) showing CXCR5 expression levels (mean fluorescence intensity, MFI) by B220^+^ msLN B cells from B6 mice (black line). FMO control (gray) included. Bar plot (**B**) reporting geometric MFI (gMFI) of CXCR5 expressed by msLN B cells from B6 and *Cxcr5*^−/−^ mice. (**C**–**H**) CXCR5 expression levels by msLN B220^−^CD11c^+^MHCII^hi^ migratory DCs from d8 *Hp*-infected B6 mice (n = 5). Migratory DC subsets (**C**) were analyzed for expression of CXCR5. CXCR5 expression levels (**D**) reported as geometric mean fluorescence intensity (gMFI) and representative histograms showing CXCR5 expression levels (MFI) by cDC1 (**E**), DN (**F**), DP (**G**) and cDC2 (**H**) cells. Bars in (**B**,**D**) denote mean ± SD of each group. Statistical analysis performed with unpaired 2-tailed Student’s *t*-test (**B**) and one-way ANOVA (**D**), **** *p* ≤ 0.001, n.s. indicates *p* > 0.05.

**Figure 3 vaccines-13-00943-f003:**
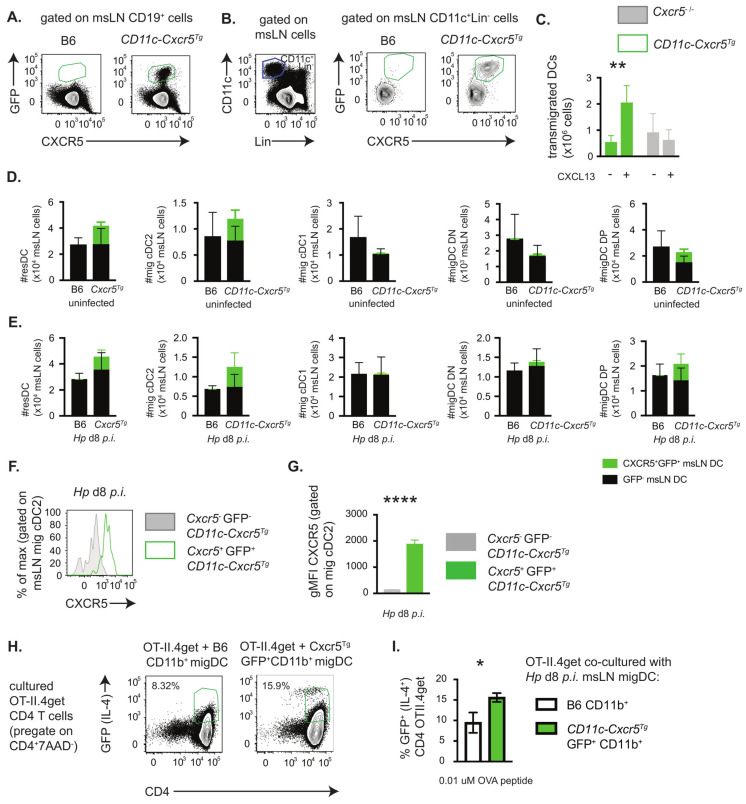
Over-expression of CXCR5 by DCs from *Hp*-infected mice enhances *in vitro* T_H_2 priming. (**A**,**B**) Expression of CXCR5 and the transgene reporter, GFP, by msLN cells from naïve and day 8 *Hp*-infected B6 and *CD11c-Cxcr5^Tg^* mice (n = 6 mice/group, see Appendix A for description of transgenic mice). Representative flow cytometry plots showing CXCR5 and GFP expression by B cells (**A**) and CD11c^+^Lin^−^ (7AAD^−^CD3^−^B220^−^NK1.1^−^Ly6G^−^Ly6C^−^CD64^−^) DCs (**B**); gating strategies for resident and migratory DCs are found in Appendix A. (**C**) CXCR5 transgene supports DC chemotaxis *in vitro*. Migration of DCs (n = 4 × 10^5^) isolated from spleens of uninfected *CD11c-Cxcr5^Tg^* and *Cxcr5*^−/−^ mice (n = 3 per genotype) was measured by transwell assay with cells placed in the top chamber and 500 ng/mL recombinant CXCL13 placed in the bottom chamber. Transmigrated cells were enumerated after 90 min. (**D**–**G**) Expression of CXCR5 and the transgene reporter, GFP, by resident and migratory DC populations from msLNs of naïve (n = 5–6/group) and day 8 *Hp*-infected (n = 6/group) *CD11c-Cxcr5^Tg^* and B6 mice. Numbers of msLN resident and migratory DC populations in uninfected (**D**) and infected (**E**) mice reported with GFP^−^ (black bars) and GFP^+^CXCR5^+^ DCs (green bars) shown. Representative histogram (**F**) showing CXCR5 expression by transgene positive (GFP^+^) and transgene negative (GFP^−^) migratory CD11b^+^CD103^−^ cDC2 cells with CXCR5 expression levels (**G**), reported as gMFI, provided. Flow plots showing CXCR5 and GFP expression by DC populations provided in Appendix A. (**H**,**I**) T_H_2 priming capacity of migratory DCs is enhanced by CXCR5 transgene expression. CD11b^+^ migratory DCs from d8 *Hp*-infected C57BL/6 mice and transgene expressing (GFP^+^) CD11b^+^ migratory DCs from d8 *Hp*-infected *CD11c-Cxcr5^Tg^* mice (n = 5 mice/group) were sort-purified (Appendix A) and co-cultured at a 1:10 ratio with splenic OTII.4get CD4 T cells in the presence of OVA peptide (0.01 mM). Representative flow plots showing expression of the IL-4 reporter by the gated CD4^+^ OTII cells (**H**) with the frequencies (**I**) of IL-4 reporter expressing (GFP^+^) CD4 T cells shown for triplicate cultures. Data representative of ≥3 independent experiments. Statistical analysis performed with unpaired 2-tailed Student’s *t*-test. Graphs show mean ± SD of each group. * *p* ≤ 0.05, ** *p* ≤ 0.01, **** *p* ≤ 0.0001.

**Figure 4 vaccines-13-00943-f004:**
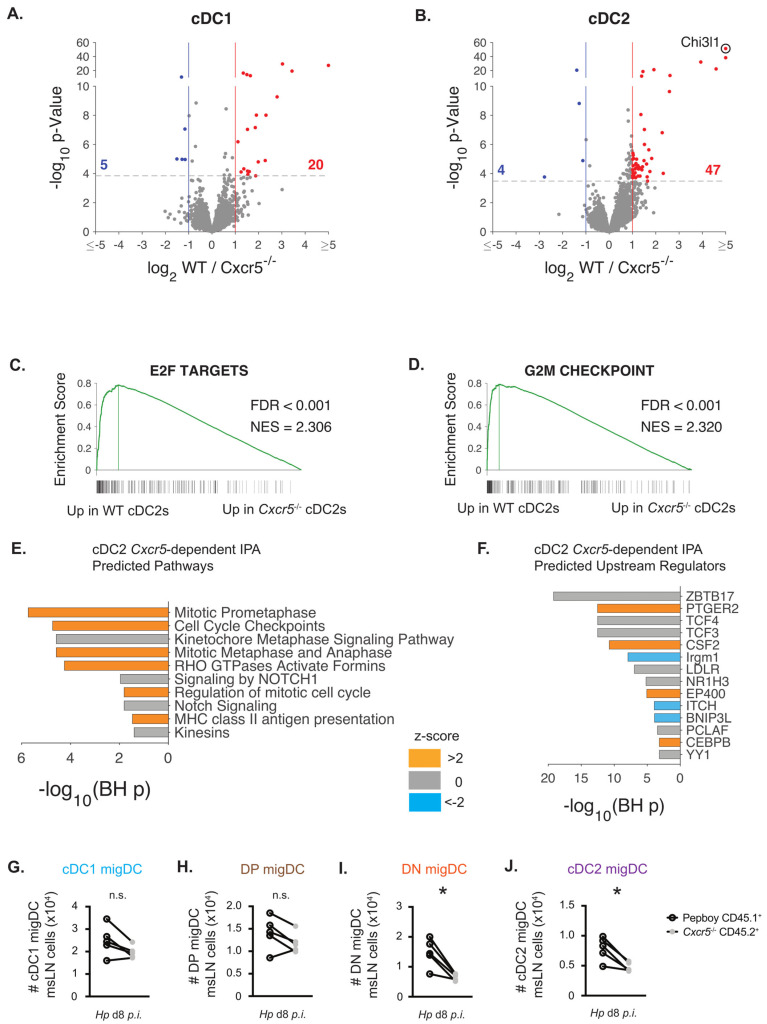
CXCR5 expression by msLN cDC2 cells promotes transcriptional proliferation programming and supports increased msLN DC recovery following *Hp* infection. (**A**–**F**) Paired RNAseq analysis of B6 and *Cxcr5*^−/−^ msLN migratory CD11b^−^CD103^+^ cDC1 cells and CD11b^+^CD103^−^ cDC2 cells isolated from *Hp*-infected 1:1 B6:*Cxcr5*^−/−^ mice. 1:1 B6:*Cxcr5*^−/−^ chimeric mice were generated (as in Figure 1A) and infected with *Hp*. B6 (CD45.1^+^) and *Cxcr5*^−/−^ (CD45.2^+^) cDC1 (CD103^+^CD11b^−^) and cDC2 (CD103^+^CD11b^+^) cells were sort-purified from individual animals as in Appendix A and bulk RNA-sequencing was performed. (**A**,**B**) DEGs (FDR *p* < 0.05 and log_2_FC of >1 or <−1) for paired analysis of WT over *Cxcr5*^−/−^ cDC1 (**A**) and cDC2 (**B**) cells shown as volcano plots. See Appendix A for complete gene expression profiles. (**C**,**D**) Gene set enrichment (GSEA) using mSigDB Hallmark gene sets to query ranked gene list of B6 cDC2 cells over *Cxcr5*^−/−^ cDC2 cells. FDRq and normalized enrichment score indicated for E2F target (**C**) and G2M checkpoint (**D**) gene sets, which are significantly enriched in B6 cDC2 cells. See Appendix A for complete Hallmark gene set data. (**E**,**F**) Ingenuity pathway (Figure 4E and Appendix A) and upstream regulator (Figure 4F and Appendix A) analyses using the 85 genes (Appendix A) meeting an FDR *p* < 0.05 cutoff when comparing gene expression between B6 and *Cxcr5*^−/−^ cDC2 cells. The −log_10_ BH corrected *p* values and activation z-scores (activated in orange, inhibited in blue, indeterminate in gray) for predicted pathways and regulators are shown. (**G**–**J**) Maintenance and/or proliferation of CD11b^+^CD103^−^ cDC2 cells and CD11b^−^CD103^−^ DN cells requires DC intrinsic expression of CXCR5. Enumeration of msLN migratory DC subsets from d8 *Hp*-infected 1:1 B6:*Cxcr5*^−/−^ chimeric mice (generated as Figure 1A). Representative gating strategy to identify B6 and *Cxcr5*^−/−^ msLN migratory cDC subpopulations in the same animal provided in Appendix A. The number of wildtype B6 CD45.1^+^ (open circles) and *Cxcr5*^−/−^ CD45.2^+^ (gray circles) msLN migratory CD103^+^CD11b^−^ cDC1 (**G**), CD103^+^CD11b^+^ DP (H), CD103^−^CD11b^−^ DN (**I**) and CD103^−^CD11b^+^ cDC2 (**J**) cells in the same animal are shown with pairs indicated by connecting line. The percentage of wildtype B6 CD45.1^+^ and *Cxcr5*^−/−^ CD45.2^+^ msLN migratory cDC1 (Appendix A), DP (Appendix A), DN (Appendix A), and cDC2 (Appendix A) cells in the same animal are reported with pairs indicted by the connecting line. Statistical analysis for RNAseq datasets (**A**–**F**) summarized in Methods. Statistical analysis in (**G**–**J**) performed with paired 2-tailed Student’s *t*-test. * *p* ≤ 0.05, n.s. indicates *p* > 0.05. Data representative of at least 3 independent experiments.

**Figure 5 vaccines-13-00943-f005:**
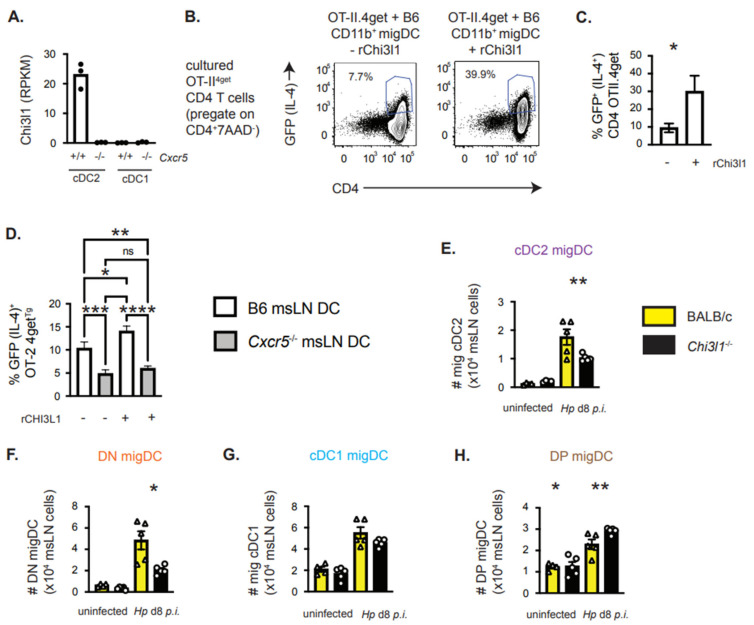
The CXCR5-dependent cDC2 gene product, Chi3l1, enhances T_H_2 priming *in vitro* and regulates the size of migratory DC subsets following *Hp* infection. (**A**) *Chi3l1* mRNA expression levels (reported as RPKM) in B6 and *Cxcr5*^−/−^ cDC1 and cDC2 cells from RNAseq analysis of *Hp*-infected 1:1 B6 Pepboy/*Cxcr5*
^−/−^ chimeric mice (see Figure 4). (**B**,**C**) Recombinant Chi3l1 protein enhances T_H_2 priming *in vitro*. CD11b^+^ CD11c^+^MHCII^hi^ migratory DCs, sort-purified (see Appendix A) from msLN of d8 *Hp*-infected B6 mice (n = 5 mice/group), were co-cultured at a 1:10 ratio with splenic OTII.4get CD4 T cells and OVA peptide (0.01 mM) in the absence or presence of 100 nM recombinant Chi3l1. Representative flow plots showing expression of the IL-4 reporter by the gated CD4^+^ OTII cells on day 4 of the cultures (**B**) with the frequencies (**C**) of IL-4 reporter expressing (GFP^+^) CD4 T cells provided for triplicate cultures. (**D**) Recombinant Chi3l1 is not sufficient to rescue defective T_H_2 priming by *Cxcr5*^−/−^ migratory DCs. CD11c-enriched msLN cells from d8 *Hp*-infected C57BL/6 and *Cxcr5*^−/−^ mice (n = 5 mice/group) were co-cultured at a 1:5 ratio with splenic OTII.4get CD4 T cells and OVA peptide (0.01 mM) in the presence or absence of 100 nM recombinant Chi3l1. Data shown as the frequencies of IL-4 reporter expressing (GFP^+^) CD4 T cells in triplicate cultures. (**E**–**H**) Enumeration of BALB/c and *Chi3l1*^−/−^ msLN migratory DC subsets in uninfected and d8 *Hp*-infected BALB/c (yellow bars) and *Chi3l1*^−/−^ (black bars) mice (n = 5 mice/group). Data reported as number of total migratory DCs (Appendix A), migratory CD103^−^CD11b^+^ cDC2 (**E**), CD103^−^CD11b^−^ DN (**F**), CD103^+^CD11b^−^ cDC1 (**G**) and CD103^+^CD11b^+^ DP (**H**) cells. Representative gating of msLN migratory DC subsets and frequencies of these subsets shown in Appendix A. Data representative of ≥3 independent experiments, displayed as the mean ± SD of each group with individual animals depicted as circles (*Chi3l1*^−/−^) or triangles (BALB/c). Statistical significance was determined using unpaired 2-tailed Student’s *t* test (**C**,**E**–**H**) or 1-way ANOVA (**D**). * *p* ≤ 0.05, ** *p* ≤ 0.01, *** *p* ≤ 0.001, **** *p* ≤ 0.0001, n.s. indicates *p* > 0.05.

**Figure 6 vaccines-13-00943-f006:**
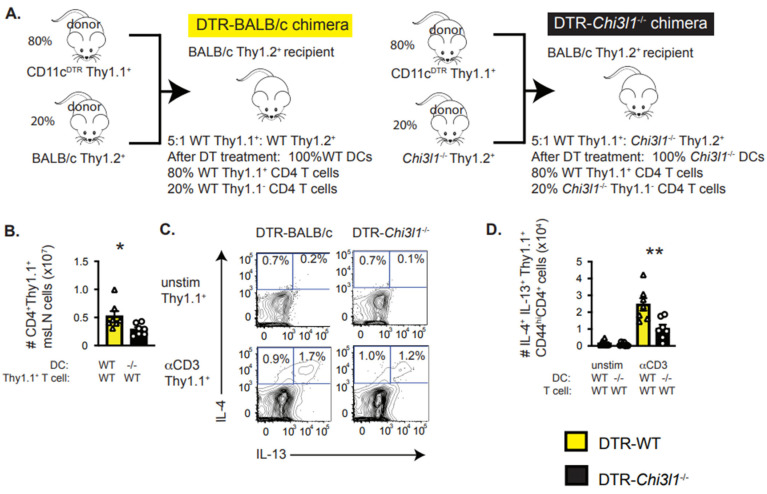
DC intrinsic expression of *Chi3l1* supports *in vivo* T_H_2 priming following *Hp* infection. (**A**) Description of BM chimeric model to test requirements for *Chi3l1* expression by DCs during T_H_2 development *in vivo*. Thy1.2^+^ BALB/c recipients were irradiated and reconstituted with a 5:1 ratio of CD11c^DTR^ Thy1.1^+^ BM cells plus BALB/c Thy1.2^+^ BM cells (DTR-BALB/c, yellow bars) or a 5:1 ratio of CD11c^DTR^ Thy1.1^+^ BM cells plus *Chi3l1*^−/−^ Thy1.2^+^ BM cells (DTR-*Chi3l1*^−/−^*,* black bars). Following 8 weeks of reconstitution, chimeric mice (n = 7group) were infected with *Hp* and exposed to 100 ng diphtheria toxin (DT) administered *i.p.* every 48 hours to eliminate DCs derived from the CD11c^DTR^ Thy1.1^+^ genotype cells, leaving either 100% WT DCs (DTR-BALB/c chimeras) or 100% *Chi3l1*^−/−^ DCs (DTR-*Chi3l1*^−/−^ chimeras). Thy1.1^+^ T cells derived from the CD11c^DTR^ Thy1.1^+^ BM precursors were all wild-type in both groups of mice and competent to express *Chi3l1*. (**B**) Enumeration of msLN Thy1.1^+^ wildtype CD4^+^ T cells in msLNs of d8 *Hp*-infected DT-treated DTR-BALB/c and DTR-*Chi3l1*^−/−^ chimeras. (**C**,**D**) Enumeration of msLN Thy1.1^+^ wildtype T_H_2 cells from msLNs of d8 *Hp*-infected DT-treated DTR-BALB/c and DTR-*Chi3l1*^−/−^ chimeras. MsLN cells from both groups of mice were isolated and restimulated *in vitro* in the presence of brefeldin A with or without plate-bound anti-CD3 antibody for 4 h, then analyzed by flow for intracellular cytokine expression. Representative flow plots (**C**) showing intracellular IL-4 and IL-13 expression by restimulated CD4^+^Thy1.1^+^CD44^hi^ msLN cells are provided and the number of CD4^+^Thy1.1^+^CD44^hi^ IL-4^+^IL-13^+^ cells (**D**) is reported. Data representative of ≥3 independent experiments, displayed as the mean ± SD of each group with individual animals depicted as circles (DTR- *Chi3l1*^−/−^) or triangles (DTR-WT). Statistical significance determined using unpaired 2-tailed Student’s *t* test. * *p* ≤ 0.05, ** *p* ≤ 0.01.

## Data Availability

All data reported in this paper will be shared by the lead contact upon request. Bulk RNA-seq data have been deposited at the Gene Expression Omnibus repository (Accession numbers GSE301250) and are publicly available from the date of publication. This paper does not report original code.

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
