# Peer review of "CXCR5 Signals Fine-Tune Dendritic Cell Transcription and Regulate TH2 Development"

_vaccines, 2025, doi:10.3390/vaccines13090943_

Round 1
Reviewer 1 Report
Comments and Suggestions for Authors
This manuscript did a very thorough study to illustrate how Cxcr5+ DCs affect Th2 development and identified a new secreted protein Chi3l1 as a critical factor. In all the authors did a great job set up the mouse models and experiments. The data is well presented. I only have a few comments below:
- Did the authors notice any physiology change of the Cxcr5-/- or Chi3l1-/- mice after Hp infection compared with wild type mice?
- Did the authors try any allergy mouse model (for example dermatitis) to test the therapeutic potential of targeting Chi3l1?
- Is there a similar requirement of Chi3l1 in human Th2 cell differentiation?
Thanks!
Author Response
Reviewer 1
This manuscript did a very thorough study to illustrate how Cxcr5+ DCs affect Th2 development and identified a new secreted protein Chi3l1 as a critical factor. In all the authors did a great job set up the mouse models and experiments. The data is well presented. I only have a few comments below:
Comment 1: Did the authors notice any physiology change of the Cxcr5-/- or Chi3l1-/- mice after Hp infection compared with wild type mice?
Response 1: We previously published manuscripts describing Hp-infected mice with a global deletion of Chi3l1 (see reference 45 in our current manuscript) and Hp-infected radiation chimeras lacking DC CXCR5 (see reference 1 in our current manuscript). We did not evaluate physiologic changes in the mice as infection with stage 3 Hp larvae induces a chronic infection in all mouse strains (including immunosufficient C56BL/6J mice). Rather, we examined whether Chi3l1 or Cxcr5 were important for establishment of the TH2, TFH and B cell responses that are known to generally contribute to anti-helminth immunity. In reference 45, we demonstrated a role for T cell and B cell intrinsic expression of Chi3l1 in regulating TFH development/survival, TH2 responses and IgE antibody production following Hp infection. In reference 1, we showed that DC-intrinsic expression of CXCR5 was required for in vivo TH2 differentiation during Hp infection. We did not study mice globally deficient in Cxcr5 as CXCR5 expression by B cells and T cells is required for establishment of the B cell follicle and optimal TFH and B cell responses. In the revised manuscript we indicate in the methods line 176-177 that we infected with Hp larvae and examined the mice at an acute timepoint before the establishment of the chronic infection with mature parasites.
Comment 2: Did the authors try any allergy mouse model (for example dermatitis) to test the therapeutic potential of targeting Chi3l1?
Response 2: We did not perform an atopic dermatitis mouse model with Chi3l1-/- mice because the essential role for Chi3l1 in atopic dermatitis has been previously published (references 49 and 51-54). Chi3l1-/- mice have also been studied in models of allergic asthma (reference 26 and 55), and food allergy (reference 50). We realized we omitted two references to atopic dermatitis publications and we have now added these references to our revised manuscript (see refs 56 and 57).
Comment 3: Is there a similar requirement of Chi3l1 in human Th2 cell differentiation?
Response 3: We are also intrigued as to whether CHI3L1 is required for human TH2 cell differentiation. While this question is beyond the scope of this manuscript, there are a number of papers (see refs 48-54) that show a strong correlation between CHI3L1 (human orthologous gene) expression and allergic disease in humans. Given the importance of TH2 responses in allergic disease, we think it likely that CHI3L1 will be important in regulating TH2 development in humans. In the future, we hope to address this question.

Reviewer 2 Report
Comments and Suggestions for Authors
In the manuscript entitled "Location dependent signals fine-tune dendritic cell transcription and regulate TH2 development" the authors study the role of CXCR5 expression (and one of the target genes, Chi3l1) on the Th2 priming/response after an Hp infection. Altough interesting data is provided, I have some major points:
- the heading is misleading as the authors do not study/show any spatial localization, any change in localization and no causal relationship to the observed effects. Pleas adapt.
- you state that only cDC2 express CXCR5, but you do not provide any data supporting this. What's about the steady-state expression of CXCR5?
- Please provide gating strategies for all subsets
- the MFIs measured on DCs are extremly low (below 200), did you perform isotype controls?
- With resprect to cDC2 how does your data realte to the heterogeneity of formerly designated cDC2 (such as [c]DC2A and [c]DC2B and, most importingly, DC3)?
- Figure 1E: How do you explain a extreme reduction in IL4 expressing cells when enhancing the Ova dose?
- the quality of Figure 3 is extremly low (very blurry). please exchange
Author Response
Reviewer 2
In the manuscript entitled "Location dependent signals fine-tune dendritic cell transcription and regulate TH2 development" the authors study the role of CXCR5 expression (and one of the target genes, Chi3l1) on the Th2 priming/response after an Hp infection. Altough interesting data is provided, I have some major points:
Comment 1: the heading is misleading as the authors do not study/show any spatial localization, any change in localization and no causal relationship to the observed effects. Pleas adapt.
Response 1: It is true that in this publication we did not show spatial localization of the cDC2 cells but in our prior publication (see ref 1), we clearly showed that CXCR5 controls the localization of the DCs in the LN following Hp infection and that localization in the perifollicular environment of the LN was critical for the function of the DCs (i.e. priming TH2 responses). Our current paper was specifically asking the question of how CXCR5, which we previously demonstrated controls localization of the DCs in the LN (ref 1), regulates TH2 priming. Thus, we felt that it was reasonable to include location in the title of the manuscript (as that is a major role for CXCR5 expression by DCs). However, we take the point that we did not provide spatial data in this manuscript. Therefore, we have changed the title to “CXCR5 signals fine-tune dendritic cell transcription and regulateTH2 development”.
Comment 2: you state that only cDC2 express CXCR5, but you do not provide any data supporting this. What's about the steady-state expression of CXCR5?
Response 2: We provided (see Figure 2D) a panel comparing the MFI of CXCR5 on cDC2 and other cDC subsets in lymph nodes of Hp-infected mice, with representative MFI and histograms for each subset shown in Figure 2E-2G. We did not assess steady-state expression of CXCR5 within DC subsets because we previously assessed this in our prior publication (see ref 1). We showed that the migratory cDC subpopulations present in the mesenteric LN (msLN) differ between steady state conditions and following Hp-infection. It is known that under steady state, msLN DC compartment predominantly consists of cDC1 and DP cDC, with very few CD11b+cD103- cDC2 migratory cDC cells. In our prior publication (ref 1) we examined the mature DCs (MHCII+CD11cint) in uninfected msLN and day 8 post-Hp infection. On day 0, the mature DCs were largely CCR7+CXCR5neg and resided within the T cell area. These mature DCs, as previously demonstrated by Brown et al. (Immunity 2019, PMID: 31668803), could spontaneously prime TH1 and TH17 cells but not TH2 cells. We showed in ref 1, that following Hp infection (but not flu infection), the mature msLN cDCs uniformly downregulated expression of CCR7 and approximately 50% of those cells upregulated expression of CXCR5. These DCs did not co-localize in the T cell zone and instead were found within the perifollicular region of the LN. We further showed that CXCR5 was required for the positioning of the migratory DCs in the LN following infection. In our current manuscript, we demonstrate that the Hp-induced migratory cDC population expressing CXCR5 is the CD11b+CD103neg subset (cDC2 cells).
Comment 3: Please provide gating strategies for all subsets
Response 3: In the prior submission, we included one representative gating strategy that was used for all experiments using DCs isolated from 1:1 C57BL/6:Cxcr5-/- mice. In the revised manuscript, we include the main gating for each experiment that is presented. The gating strategies for all figures are as follows:
Figure 1B in revised Figure 1, includes the gating that was used for the experiments presented in Figure 1D-E. The gating strategy shown in Figure 1B is also identical to that used to define the migratory DC subset (MHCIIhiCD11cint).
Figure 2C gating strategy is shown for the cDC subset panels in Figure 2E-G; the upstream migratory DC gate is analogous to that shown in Figure 1B.
Figure 3B migratory DC gating is added to Figure 3B (described in legend line 456). Gating for Figures 3D-F are shown in supplemental Figure S1C-F (described in legend line 466).
Figure 4A-F DC gating for RNAseq input is shown in Figure S2A-B. Gating for Figure 4G-J is shown in Figure S2D-M. For consistency and increased readability we have modified the representative flow panels in Figure 1 and Figure S2 to use similar flow panel styles (contour graphs and dot plots).
Gating for Figure 5E-H is shown Figure S3C as described in line 617 of the legend.
Comment 4: the MFIs measured on DCs are extremly low (below 200), did you perform isotype controls?
Response 4: We did not perform isotype control staining for CXCR5 in these experiments. However, we validated our antibody using the Cxcr5-/- B cells in Figure 2B. Since B cells express much higher levels of CXCR5 than DCs, we felt that this allowed us to accurately set a threshold between real staining and background. As an additional control, we analyzed CXCR5 staining by the cDC2 cells from 1:1 B6:Cxcr5-/- mice. These data, which show low but detectable CXCR5 expression by the B6 cDC2 cells, is now provided in supplemental Figure S2C
Comment 5: With resprect to cDC2 how does your data realte to the heterogeneity of formerly designated cDC2 (such as [c]DC2A and [c]DC2B and, most importingly, DC3)?
Response 5: We did not perform an analysis for surface markers of cDC2A, cDC2B and DC3 during our experiments. Therefore, we cannot comment on CXCR5 expression as a hallmark of these subsets or the role of these subsets in TH2 priming during Hp infection. In the future, it will be important to assess whether CXCR5 expression is heterogenous within the subpopulations that are found within the cDC2 compartment.
Comment 6: Figure 1E: How do you explain a extreme reduction in IL4 expressing cells when enhancing the Ova dose?
Response 6: Enhanced TH2 priming at low doses of antigen was first described in 1997 by Dr. Kim Bottomly (Boutin et al. J Immunol 1997, PMID: 9550376), and Dr. William Paul identified that dendritic cells spontaneously promote TH2 priming only in the setting of low peptide concentrations in 2005 (Yamane et al. J Exp Med 2005, PMID: 16172258). An enlightening review of this topic by Dr. Paul was written in 2010 (Paul Immunol Cell Biol 2010, PMID: 20157328) and it is the subject of several reviews on dendritic cell TH2 priming mechanisms including those referenced in our manuscript. Thus, what we found is consistent with the published literature and our understanding of how early TCR and co-stimulatory signals support TH2 development.
Comment 7: the quality of Figure 3 is extremly low (very blurry). please exchange.
Response 7: We apologize for this. We have now uploaded a higher-quality version of Figure in the revised manuscript. We hope that this has solved the problem.

Reviewer 3 Report
Comments and Suggestions for Authors
The Authors conducted a robust experiment to determine the role of CXCR5 in the process of Th2 cells priming by DCs. The number of mice with different genetic backgrounds is used. The paper is well written; nevertheless, there are some issues to be addressed. The author uses multiple techniques and describes them well. Although professional language is used (this is an advantage), some fragments need simplification or additional explanation since the paper will not only be read by the data analysis scientists or cytometer specialists. Classical parasitologists and immunologists may also be interested in the MS. The scientific community is very diverse.
- The Authors clearly stated in the Abstract that:
“To understand how CXCR5 facilitates the TH2 priming capabilities of migratory cDC2 cells, we performed RNAseq analysis on wildtype and Cxcr5-/- DC subset” (28-29).
The description in the MS reflects this:
"RNA was isolated (RNeasy micro column (Qiagen)) from sorted msLN migratory DCs of Hp-infected 1:1 B6:Cxcr5-/- mice (described above" (258).
And the Fig. 4 caption.
“Paired RNAseq analysis of B6 and Cxcr5-/- msLN migratory CD11b-CD103+ cDC1 cells and CD11b+CD103- cDC2 cells isolated from Hp-infected 1:1 B6:Cxcr5-/- mice. “(520 -521)
Please specify that the results show the genes that were differentially expressed between the two populations. It will be better for the general audience.
- The introduction should be enriched in the DC subsets classification. DCs can be divided into conventional and plasmacytoid. Migratory DCs belong to the conventional population. Or maybe there is some other classification I am not aware of.
- The abbreviation used in the abstract mentioned for the first time should be explained rapidly for the Reader`s convenience, e.g., “We showed that migratory conventional DC2 (cDC2) cells express…”
Author Response
Reviewer 3
The Authors conducted a robust experiment to determine the role of CXCR5 in the process ofTh2 cells priming by DCs. The number of mice with different genetic backgrounds is used. The paper is well written; nevertheless, there are some issues to be addressed. The author uses multiple techniques and describes them well. Although professional language is used(this is an advantage), some fragments need simplification or additional explanation since the paper will not only be read by the data analysis scientists or cytometer specialists. Classical parasitologists and immunologists may also be interested in the MS. The scientific community is very diverse.
Comment 1: The Authors clearly stated in the Abstract that: “To understand how CXCR5 facilitates the TH2 priming capabilities of migratory cDC2 cells, we performed RNAseq analysis on wildtype and Cxcr5-/- DC subset” (28-29). The description in the MS reflects this: “RNA was isolated (RNeasy micro column (Quiagen)) from sorted msLN migratory DCs of Hp-infected 1:1 B6;Cxcr5-/- mice (described above” (258). And the Fig 4. caption. “Paired RNAseq analysis of B6 and Cxcr5-/- msLN migratory CD11b-CD103+ cDC1 cells and CD11b+CD103- cDC2 cells isolated from Hp-infected 1:1 B6:Cxcr5-/- mice. “(520 -520) Please specify that the results show the genes that were differentially expressed between the two populations. It will be better for the general audience.
Response 1: In the revised manuscript, we added to the Figure 4 caption title (lines 519-520) and text (lines 520-523) the terminology “differential gene expression”.
Comment 2: The introduction should be enriched in the DC subsets classification. DCs can be divided into conventional and plasmacytoid. Migratory DCs belong to the conventional population. Or maybe there is some other classification I am not aware of.
Response 2: We have added to the introduction the delineation of conventional and plasmacytoid dendritic cells, and that conventional DCs within draining lymph nodes are then described as “migratory” or “resident” cells. Please see lines 52-55 of the revised manuscript.
Comment 3: The abbreviation used in the abstract mentioned for the first time should be explained rapidly for the Reader`s convenience, e.g., “We showed that migratory conventional DC2 (cDC2) cells express…”
Response 3: We have added full terms with abbreviations to the abstract for the following: cDC2 (line 26-27), TH2 (line 24), and Chi3l1 (line 35) of the revised manuscript.

Round 2
Reviewer 2 Report
Comments and Suggestions for Authors
Thank you, all my concerns have been addressed.